# Statistical inferences for type-II hybrid censoring data from the alpha power exponential distribution

**Mukhtar M. Salah**[1], **Essam A. Ahmed**[2]*, **Ziyad A. Alhussain**[1], **Hanan Haj Ahmed**[3], **M. El-Morshedy**[4,5], **M. S. Eliwa**[1,5]

**1** Department of Mathematics, College of Science in Al-Zulfi, Majmaah University, Majmaah, Saudi Arabia, **2** Mathematics Department, Sohag University, Sohag, Egypt, **3** Department of Basic Science, Preparatory Year Deanship, King Faisal University, Hofuf, Al-Ahsa, Saudi Arabia, **4** Department of Mathematics, College of Science and Humanities in Al-Kharj, Prince Sattam bin Abdulaziz University, AlKharj, Saudi Arabia, **5** Department of Mathematics, Faculty of Sciences, Mansoura University, Mansoura, Egypt

* essam.mohamed@science.sohag.edu.eg

## Abstract

This paper describes a method for computing estimates for the location parameter $\mu > 0$ and scale parameter $\lambda > 0$ with fixed shape parameter $\alpha$ of the alpha power exponential distribution (APED) under type-II hybrid censored (T-IIHC) samples. We compute the maximum likelihood estimations (MLEs) of $(\mu, \lambda)$ by applying the Newton-Raphson method (NRM) and expectation maximization algorithm (EMA). In addition, the estimate hazard functions and reliability are evaluated by applying the invariance property of MLEs. We calculate the Fisher information matrix (FIM) by applying the missing information rule, which is important in finding the asymptotic confidence interval. Finally, the different proposed estimation methods are compared in simulation studies. A simulation example and real data example are analyzed to illustrate our estimation methods.

## 1 Introduction

In experiments involving life testing and reliability, complete failure time information may not be achieved for all items. Therefore, the data obtained from this type of experiment are called censored data, for which cost effectiveness and minimization of the total testing time are important. Various censoring schemes that can be used in reliability analysis have been published. The two most frequently considered censoring schemes in the reliability literature are the type-I censoring scheme (T-ICS) and T-IICS. In T-ICS, a test stops at a prefixed time $T$, whereas in T-IICS, the test stops after a prefixed $r$ test units have failed. Thus, the number of observed failures is random in T-ICS and the duration of the experiment is random in T-IICS.

T-ICS and T-IICS can be combined to form a hybrid censoring scheme (HCS), as first presented by Epstein [1], who studied the properties of the one-parameter exponential distribution. For more details about HCS, the reader is referred to Balakrishnan and Kundu [2]. HCSs are categorized as type-I and type-II. To see this, a type-I hybrid censoring scheme (T-IHC) is defines as follows. If the test is stopped at a random time, $T^* = \min\{X_{r:n}, T\}$, where $r$ and $T$ are

**Data Availability Statement:** All relevant data are within the manuscript.

**Funding:** The authors extend their appreciation to the Deanship of Scientific Research at Majmaah

University for funding this work under project number No. (RGP-2019-2).

Competing interests: The authors have declared that no competing interests exist.

prefixed numbers, $n$ is the sample size, and $X_{r:n}$ indicates the time of the $r$th failure. In many papers, the T-IHC was examined, for example Kundu [3], and Kundu and Pradhan [4]. If the test is stopped at time $T^* = \max\{X_{r:n}, T\}$, the HCS is called T-IIHC [5]. The advantage of this approach is that it allows the complete lifetimes of at least $r$ units to be recorded before the experiment is stopped. Several authors, such as Banerjee and Kundu [6], Balakrishnan and Shafay [7], Singh et al. [8], and Salah [9–11], considered statistical inference under T-IIHC. Dey et al. [12] studied the estimation of the generalized inverted exponential distribution under a HCS. The weighted exponential distribution was considered for the T-IIHC hybrid in Kohansal et al. [13]. Singh et al. [14] presented an estimation procedure for the two-parameter Lomax distribution under a T-IIHC. They considered applying MLEs and Bayes' estimation for parameter and reliability studies, see Dong et al. [15]. Sharma [16] discussed an estimation procedure for sample prediction problems based on a T-IIHC sample for the flexible Weibull distribution. Recently, Sen et al. [17] studied the T-IIHC for the generalized exponential distribution.

In recent years, many distributions have been generalized by adding more shape parameters because many applications in engineering, finance, biomedicine, and environmental science indicated that such distributions are not powerful for explaining data sets. Hence, to make effective progress in these applications, continuous extension of these distributions is required. More recently, Mahdavi and Kundu [18] presented a novel method, called alpha power transformation (APT), for adding an extra parameter to a continuous distribution. The proposed method is useful for incorporating skewness in a family of distributions. They applied their method to the one-parameter exponential distribution and produced the two-parameter APED. Additionally, they introduced many properties of the APED, such as explicit expressions for the order statistics, moment functions and quantiles. The probability density function (PDF), and the cumulative distribution function (CDF) of the APED are, respectively,

$$
f(x; \mu, \lambda) = \begin{cases} \dfrac{\ln\alpha}{\alpha - 1} \lambda e^{-\lambda(x-\mu)} \alpha^{1 - e^{-\lambda(x-\mu)}} & \text{if } \alpha \neq 1, \\ \lambda e^{-\lambda(x-\mu)} & \text{if } \alpha = 1, \end{cases}
\tag{1}
$$

$$
F(x; \mu, \lambda) = \begin{cases} \dfrac{\alpha^{1 - e^{-\lambda(x-\mu)}} - 1}{\alpha - 1} & \text{if } \alpha \neq 1, \\ 1 - e^{-\lambda(x-\mu)} & \text{if } \alpha = 1, \end{cases}
\tag{2}
$$

The corresponding reliability and hazard rate functions are

$$
R(t) = \begin{cases} \dfrac{\alpha - \alpha^{1 - e^{-\lambda(x-\mu)}}}{\alpha - 1} & \text{if } \alpha \neq 1, \\ e^{-\lambda(x-\mu)} & \text{if } \alpha = 1, \end{cases}
\tag{3}
$$

and

$$
H(t) = \begin{cases} \dfrac{\lambda e^{-\lambda(t-\mu)} \alpha^{1 - e^{-\lambda(t-\mu)}} \ln\alpha}{\alpha^{1 - e^{-\lambda(t-\mu)}} - 1} & \text{if } \alpha \neq 1, \\ \dfrac{\lambda e^{-\lambda(t-\mu)}}{1 - e^{-\lambda(t-\mu)}} & \text{if } \alpha = 1, \end{cases}
\tag{4}
$$

where $X > 0$, $\mu > 0$ and $\lambda > 0$. Note that in the rest of this paper, it is assumed that $\alpha \neq 1$. For more details about APED see Salah [19].

The primary purpose of the present paper is to propose an estimation method for the parameters and the reliability characteristics for APED using incomplete sample observations obtained by a T-IIHC. To the best of our knowledge, no attempt has been made to estimate these characteristics for APED using a T-IIHC. This study aims to fill this gap using MLEs via EMA to compute and compare the outcomes with those calculated using the NRM. Furthermore, the asymptotic confidence intervals (ACIs) for the APED parameters were computed. The rest of this article is organized as follows. In the following section, the MLEs of the unknown parameters and the failure functions and reliability are discussed. The ACIs and the FIM are presented in Sections 3 and 4, respectively. A real data set is examined for illustrative purposes in Section 5, while conclusions are discussed in Section 6.

## 2 Maximum likelihood estimation

MLE is an important and widely used method for fitting statistical models because of its attractive properties, such as asymptotic efficiency, consistency, and asymptotic unbiasedness. Here, we describe the attainment of MLEs of the model parameters based on T-IIHC samples via the NRM and EMA.

### 2.1 Newton–Raphson algorithm

Consider a sample of $n$ units and let $T$ be a preselected experimental time and $r$ a predetermined number of units out of a total of $n$ units. The experiment ends at time $T^* = \max\{x_{k:n}, T\}$. Furthermore, let $J$ be the number of failures that occurred before time $T$, i.e, $J = \sum_{i=1}^{n} I\{X_i < T\}$.

Then, under the T-IIHC, we have the following observations:

**Case 1**: $x_{1:n} < x_{2:n} < \ldots < x_{r:n}$ if $x_{r:n} > T$, or
**Case 2**: $x_{1:n} < \ldots < x_{r:n} < x_{r+1:n} < x_{J:n} < T < x_{J+1:n}$ if $r \leq J < n$, and $x_{J:n} < T < x_{J+1:n}$, or
**Case 3**: $x_{1:n} < x_{2:n} < \ldots < x_{n:n} < T$, if $x_{n:n} < T$.
The likelihood functions of these three different cases are:

$$L(\mu, \lambda|x) = \begin{cases} \dfrac{n!}{(n-r)!} \displaystyle\prod_{i=1}^{r} f(x_i)[1 - F(x_r)]^{(n-r)} & \text{if } x_{r:n} > T \\[3ex] \dfrac{n!}{(n-J)!} \displaystyle\prod_{i=1}^{J} f(x_i)[1 - F(T)]^{(n-J)} & \text{if } r < J < n, \text{ and } x_{J:n} < T < x_{J+1:n} \\[3ex] n! \displaystyle\prod_{i=1}^{n} f(x_i) & \text{if } x_{n:n} < T. \end{cases}$$

Hence,

$$L(\mu, \lambda|x) = \begin{cases} \dfrac{n!}{(n-r)!} \displaystyle\prod_{i=1}^{r} \dfrac{\lambda \ln \alpha}{\alpha - 1} e^{-\lambda(x_i - \mu)} \alpha^{1 - e^{-\lambda(x_i - \mu)}} \left[ \dfrac{\alpha - \alpha^{1 - e^{-\lambda(x_r - \mu)}}}{\alpha - 1} \right]^{(n-r)} & \text{if } x_{r:n} > T, \\[3ex] \dfrac{n!}{(n-J)!} \displaystyle\prod_{i=1}^{J} \dfrac{\lambda \ln \alpha}{\alpha - 1} e^{-\lambda(x_i - \mu)} \alpha^{1 - e^{-\lambda(x_i - \mu)}} \left[ \dfrac{\alpha - \alpha^{1 - e^{-\lambda(T - \mu)}}}{\alpha - 1} \right]^{(n-J)} & \text{if } r < J < n, \\[3ex] n! \displaystyle\prod_{i=1}^{n} \dfrac{\lambda \ln \alpha}{\alpha - 1} e^{-\lambda(x_i - \mu)} \alpha^{1 - e^{-\lambda(x_i - \mu)}}. \end{cases} \tag{5}$$

By combining the three likelihood functions, one obtains

$$L(\mu, \lambda | x) = \frac{n!}{(n-k)!} \prod_{i=1}^{k} \frac{\lambda \ln \alpha}{\alpha - 1} e^{-\lambda(x_i - \mu)} \alpha^{1 - e^{-\lambda(x_i - \mu)}} \left[ \frac{\alpha - \alpha^{1 - e^{-\lambda(c-\mu)}}}{\alpha - 1} \right]^{(n-k)},$$

where $k$ and $c$ are given by

$$(k, c) = \begin{cases} (r, x_r) & \text{if } x_{r:n} > T, \\ (J, T) & \text{if } x_{n:n} < T, J > r, \\ (n, T) & \text{if } x_{n:n} < T. \end{cases}$$

The log-likelihood function $\ell = \ln L(\mu, \lambda)$ without the constant term can be written as

$$\ell = k \ln [\lambda] - \lambda \sum_{i=1}^{k} (x_i - \mu) - \ln \alpha \sum_{i=1}^{k} e^{-\lambda(x_i - \mu)} + (n-k) \ln \left[ \frac{\alpha - \alpha^{1 - e^{-\lambda(c-\mu)}}}{\alpha - 1} \right]. \quad (6)$$

The MLEs of $\mu$ and $\lambda$ are obtained by differentiating Eq (6) with respect to $\mu$ and $\lambda$. The simultaneous equations are expressed as

$$\frac{\partial \ell}{\partial \mu} = k\lambda - \lambda \ln \alpha \sum_{i=1}^{k} e^{-\lambda(x_i - \mu)} + (n-k) \frac{\lambda e^{-\lambda(c-\mu)} \ln \alpha}{\alpha^{e^{-\lambda(c-\mu)}} - 1} = 0, \quad (7)$$

and

$$\frac{\partial \ell}{\partial \lambda} = \frac{k}{\lambda} - \sum_{i=1}^{k} (x_i - \mu) + \ln \alpha \sum_{i=1}^{k} (x_i - \mu) e^{-\lambda(x_i - \mu)} - \frac{(n-k)(c-\mu) e^{-\lambda(c-\mu)} \ln \alpha}{\alpha^{e^{-\lambda(c-\mu)}} - 1} = 0. \quad (8)$$

It is difficult to obtain an analytical solution to these nonlinear equations. Therefore, we can estimate the parameters $\mu$ and $\lambda$ using statistical software or by solving the two simultaneous Eqs (7) and (8) numerically by, for example, NRM with a good initial guess of $\mu^{(0)}$ and $\lambda^{(0)}$.

Utilizing the property of invariance (replacing $\mu$ and $\lambda$ by their ML estimators $\hat{\mu}_{ML}$ and $\hat{\lambda}_{ML}$), we can obtain the MLE of the reliability and hazard function from Eqs (3) and (4) by

$$\hat{R}_{ML}(t) = \frac{\alpha - \alpha^{1 - e^{-\hat{\lambda}_{ML}(t - \hat{\mu}_{ML})}}}{\alpha - 1} \quad (9)$$

and

$$\hat{H}_{ML}(t) = \frac{\hat{\lambda}_{ML} e^{-\hat{\lambda}_{ML}(t - \hat{\mu}_{ML})} \alpha^{1 - e^{-\hat{\lambda}_{ML}(t - \hat{\mu}_{ML})}} \ln \alpha}{\alpha^{1 - e^{-\hat{\lambda}_{ML}(t - \hat{\mu}_{ML})}} - 1}. \quad (10)$$

## 2.2 Expectation maximization algorithm

The EMA is a very powerful method for finding MLEs for parametric models when the data are censored; see Dempster et al. [20]. It consists of two iterative steps: (i) the expectation step (E-step) and (ii) the maximization step (M-step). The E-step of each iteration computes only the conditional expectation of the log-likelihood with respect to the incomplete data given the observed data. In the M-step, the parameter value is calculated by maximizing the expected log-likelihood function obtained in the E-step. For more details about the EMA, see McLachlan and Krishnan [21].

Let $X = (X_{1:n}, X_{2:n}, \ldots, X_{k:n})$ denote the observed data and $Z = (Z_1, Z_2, \ldots, Z_{n-k})$ denote the censored data. Here, for a given $k$, $Z_1, Z_2, \ldots, Z_{n-k}$ are not observable. The complete data are given by the combination of $W = (X, Z)$.

In the E-step, the conditional expected value of the log-likelihood for the given complete, observed sample must be calculated. Hence, the log-likelihood function for the complete data is

$$L_c(\mu, \lambda; W) = n \ln \lambda - \lambda \sum_{i=1}^{k}(x_i - \mu) - \lambda \sum_{i=1}^{n-k}(z_i - \mu) - \ln \alpha \sum_{i=1}^{k} e^{-\lambda(x_i - \mu)} - \ln \alpha \sum_{i=1}^{n-k} e^{-\lambda(z_i - \mu)}, \quad (11)$$

and

$$\begin{aligned} L_s(\mu, \lambda) &= n \ln \lambda - \lambda \left\{ \sum_{i=1}^{k}(x_i - \mu) + \sum_{i=1}^{n-k} E[(z_i - \mu)|z_i > c] \right\} \\ &\quad - \ln \alpha \left\{ \sum_{i=1}^{k} e^{-\lambda(x_i - \mu)} + \sum_{i=1}^{n-k} E[e^{-\lambda(z_i - \mu)}|z_i > c] \right\}. \end{aligned} \quad (12)$$

Now, for $j = 1, \ldots, n-k$, the PDF of $Z_j$ given $X_{1:n} = x_{1:n}, X_{2:n} = x_{2:n}, \ldots, X_{k:n} = x_{k:n}$ is given by (see Ng et al. [22])

$$f_{Z|X}\left(z_j|X\right) = \frac{f(z_j; \mu, \lambda)}{1 - F(c, \mu, \lambda)} = \frac{\ln(\alpha)\lambda e^{-\lambda(z_j - \mu)} \alpha^{1 - e^{-\lambda(z_j - \mu)}}}{(\alpha - 1)(1 - F(c, \mu, \lambda))}, z_j > c. \quad (13)$$

According to Eq (13), we can write

$$A(c; \mu, \lambda) = E[(z_i - \mu)|Z_i > c)] = \frac{\ln \alpha}{-\lambda(\alpha - 1)(1 - F(c, \mu, \lambda))} \int_{0}^{e^{-\lambda(c-\mu)}} \ln(u) \alpha^{1-u} du, \quad (14)$$

and

$$B(c; \mu, \lambda) = E[e^{-\lambda(z_i - \mu)}|Z_i > c] = \frac{\alpha - \alpha^{1 - e^{-\lambda(c-\mu)}}(1 + e^{-\lambda(c-\mu)} \ln \alpha)}{(\alpha - 1)(1 - F(c, \mu, \lambda)) \ln \alpha}. \quad (15)$$

Next, the M-step involves the maximization of Eq (12); if at the $h$-th stage, the estimate of $(\mu, \lambda)$ is $(\mu^{(h)}, \lambda^{(h)})$, then $(\mu^{(h+1)}, \lambda^{(h+1)})$ can be estimated by maximizing

$$\begin{aligned} \Phi(\mu, \lambda|\mu^{(h)}, \lambda^{(h)}) &= n \ln \lambda - \lambda \left\{ \sum_{i=1}^{k}(x_i - \mu) + (n - k)A(c; \mu^{(h)}, \lambda^{(h)}) \right\} \\ &\quad - \ln \alpha \left\{ \sum_{i=1}^{k} e^{-\lambda(x_i - \mu)} + (n - k)B(c; \mu^{(h)}, \lambda^{(h)}) \right\}. \end{aligned} \quad (16)$$

By taking the derivative of Eq (16) w.r.t $\mu$ and $\lambda$ and setting them equal to 0, we first find $\lambda^{(h+1)}$ by solving

$$g(\lambda) = \lambda$$

and $g(\lambda)$ is given as follows

$$g(\lambda) = \left[ \frac{1}{n} \sum_{i=1}^{k}(x_i - \hat{\mu}(\lambda)) - \frac{1}{n} \sum_{i=1}^{k}(x_i - \hat{\mu}(\lambda)) e^{-\lambda(x_i - \hat{\mu}(\lambda))} \ln \alpha + \frac{1}{n}(n - k)A(c, \mu^{(h)}, \lambda^{(h)}) \right]^{-1} \quad (17)$$

where

$$\hat{\mu}_{EMA}(\lambda) = \frac{1}{\lambda} \ln \left[ k \left( \ln \alpha \sum_{i=1}^{k} e^{-\lambda x_i} \right)^{-1} \right].$$

(18)

Then, $\mu^{(h+1)}$ is obtained as $\mu^{(h+1)} = \hat{\mu}_{EMA}(\lambda^{(h+1)})$.

**Remark**: The iterative scheme for obtaining the MLEs of $(\mu, \lambda)$ using the EMA is terminated when $|\mu^{(h+1)} - \mu^{(h)}| + |\lambda^{(h+1)} - \lambda^{(h)}| < \epsilon$, where $\epsilon > 0$ is a preassigned small number. When convergence occurs, the present $\mu^{(h+1)}$ and $\lambda^{(h+1)}$ are the MLEs of $\mu$ and $\lambda$ obtained via the EMA; we refer to these values as $(\hat{\mu}_{EMA}, \hat{\lambda}_{EMA})$.

According to the invariant property of MLEs, the MLEs of the reliability and hazard functions of APED via the EMA, denoted by $\hat{R}_{EMA}(t)$ and $\hat{H}_{EMA}(t)$, respectively, can be obtained by replacing $\mu$ and $\lambda$ in Eqs (3) and (4) with their MLE estimates

$$\hat{R}_{EMA}(t) = \frac{\alpha - \alpha^{1-e^{-\hat{\lambda}_{EMA}(t-\hat{\mu}_{EMA})}}}{\alpha - 1}$$

(19)

and

$$\hat{H}_{EMA}(t) = \frac{\hat{\lambda}_{EMA} e^{-\hat{\lambda}_{EMA}(t-\hat{\mu}_{EMA})} \alpha^{1-e^{-\hat{\lambda}_{EMA}(t-\hat{\mu}_{EMA})}} \ln \alpha}{\alpha^{1-e^{-\hat{\lambda}_{EMA}(t-\hat{\mu}_{EMA})}} - 1}.$$

(20)

In the following section, we consider interval estimation and the reliability and hazard functions of APED$(\mu, \lambda)$ under type-II hybrid censored data.

## 3 Confidence intervals

### 3.1 Asymptotic confidence intervals

In this subsection, we derive the ACIs of $\mu$, $\lambda$, $R(t)$, and $H(t)$. To achieve this aim, we use the bivariate central limit theorem to obtain the asymptotic distribution of the unknown parameters, i.e., $\mu$ and $\lambda$, and apply the delta method to determine the asymptotic distributions of $R(t)$ and $H(t)$.

One of the characteristics that distinguishes the MLE is its asymptotic variance of the inverse of the Fisher information matrix. Because the MLEs of the parameters are not obtained in a closed form, it is not possible to obtain the Fisher information matrix and construct ACIs. Therefore, the Fisher information is approximated using the observed Fisher information evaluated at the MLE. The ACIs based on the asymptotic normal distribution of the MLEs are approximated as the inverse of the observed Fisher information matrix evaluated at the MLE.

The two unknown parameters $\mu$ and $\lambda$ are approximately bivariate normal with mean $(\hat{\mu}_{ML}, \hat{\lambda}_{ML})$ and variance–covariance matrix $I(\hat{\mu}, \hat{\lambda})$, (see Greene [23] and Agresti [24]), where

$$I(\hat{\mu}_{ML}, \hat{\lambda}_{ML}) = -\begin{bmatrix} \dfrac{\partial^2 \ell}{\partial \mu^2} & \dfrac{\partial^2 \ell}{\partial \mu \partial \lambda} \\[2mm] \dfrac{\partial^2 \ell}{\partial \mu \partial \lambda} & \dfrac{\partial^2 \ell}{\partial \lambda^2} \end{bmatrix}_{(\mu,\lambda)=(\hat{\mu}_{ML}, \hat{\lambda}_{ML})}, \qquad (21)$$

$$\frac{\partial^2 \ell}{\partial \mu^2} = -\lambda^2 \ln \alpha \sum_{i=1}^{k} e^{-\lambda(x_i-\mu)} - \frac{(n-k)\lambda^2 e^{-\lambda(c-\mu)} \ln \alpha [1 - \alpha^{e^{-\lambda(c-\mu)}} (1 - e^{-\lambda(c-\mu)} \ln \alpha)]}{(\alpha^{e^{-\lambda(c-\mu)}} - 1)^2},$$

$$\begin{aligned} \frac{\partial^2 \ell}{\partial \mu \partial \lambda} &= \frac{\partial^2 \ell}{\partial \lambda \partial \mu} = k + \ln \alpha \sum_{i=1}^{k} e^{-\lambda(x_i-\mu)} [\lambda(x_i - \mu) - 1] + \\ &\quad \frac{(n-k)e^{-\lambda(c-\mu)} \ln \alpha [(\alpha^{e^{-\lambda(c-\mu)}} - 1)(\lambda(c-\mu) - 1) - \lambda \ln \alpha(c-\mu)e^{-\lambda(c-\mu)}\alpha^{e^{-\lambda(c-\mu)}}]}{(\alpha^{e^{-\lambda(c-\mu)}} - 1)^2}, \end{aligned}$$

$$\begin{aligned} \frac{\partial^2 \ell}{\partial \lambda^2} &= -\frac{1}{\lambda^2} - \ln \alpha \sum_{i=1}^{k} (x_i - \mu)^2 e^{-\lambda(x_i-\mu)} - \\ &\quad \frac{(n-k)(c-\mu)^2 e^{-\lambda(c-\mu)} \ln \alpha [1 - \alpha^{e^{-\lambda(c-\mu)}} (1 - e^{-\lambda(c-\mu)} \ln \alpha)]}{(\alpha^{e^{-\lambda(c-\mu)}} - 1)^2}, \end{aligned}$$

is the inverse of the matrix in Eq (21). The variance–covariance matrix is then

$$I^{-1}(\hat{\mu}_{ML}, \hat{\lambda}_{ML}) = \begin{bmatrix} \widehat{Var}(\hat{\mu}_{ML}) & Cov(\hat{\mu}_{ML}, \hat{\lambda}_{ML}) \\[2mm] Cov(\hat{\mu}_{ML}, \hat{\lambda}_{ML}) & \widehat{Var}(\hat{\lambda}_{ML}) \end{bmatrix}. \qquad (22)$$

Therefore, the large sample theorem can be used to compute the two-sided $100(1 - \gamma)\%$ estimated confidence intervals for $\mu$ and $\lambda$, respectively, as

$$\hat{\mu}_{ML} \pm Z_{\frac{\gamma}{2}} \sqrt{\widehat{Var}(\hat{\mu}_{ML})} \text{ and } \hat{\lambda}_{ML} \pm Z_{\frac{\gamma}{2}} \sqrt{\widehat{Var}(\hat{\lambda}_{ML})}, \qquad (23)$$

where $Z_{\frac{\gamma}{2}}$ is the percentile of the standard normal distribution with right-tail probability $\gamma/2$. Moreover, to construct the ACIs for the reliability and hazard functions, we apply the delta method to estimate their variances. Let

$$G_1^T = \left( \frac{\partial R(t)}{\partial \mu} \frac{\partial R(t)}{\partial \lambda} \right), G_2^T = \left( \frac{\partial H(t)}{\partial \mu} \frac{\partial H(t)}{\partial \lambda} \right). \qquad (24)$$

Then, the asymptotic estimators of $Var(\hat{R})$ and $Var(\hat{H})$ are defined as

$$\widehat{Var}(\hat{R}) = [G_1^T I^{-1} G_1]_{(\mu,\lambda)=(\hat{\mu}_{ML}, \hat{\lambda}_{ML})}, \widehat{Var}(\hat{H}) = [G_2^T I^{-1} G_2]_{(\mu,\lambda)=(\hat{\mu}_{ML}, \hat{\lambda}_{ML})}, \qquad (25)$$

where $I^{-1}$ is the variance–covariance matrix defined in Eq (22). Therefore, we have the relationships

$$\frac{\hat{R}_{ML}(t) - R(t)}{\sqrt{\widehat{Var}(\hat{R})}} \sim N(0, 1), \frac{\hat{H}_{ML}(t) - H(t)}{\sqrt{\widehat{Var}(\hat{H})}} \sim N(0, 1)$$

Furthermore, we can derive the $100(1 - \gamma)\%$ ACIs of $R(t)$ and $H(t)$ by

$$\hat{R}_{ML}(t) \pm Z_{\frac{\gamma}{2}}\sqrt{\widehat{Var}(\hat{R})} \text{ and } \hat{H}_{ML}(t) \pm Z_{\frac{\gamma}{2}}\sqrt{\widehat{Var}(\hat{H})}. \qquad (26)$$

### 3.2 Fisher information matrix

This section presents the observed FIM using the missing value rule of Louis [25]. To construct the ACIs, as

$$Observed\ information = Complete\ information - Missing\ information.$$

$$I_X(\theta) = I_W(\theta) - I_{W|X}(\theta), \qquad (27)$$

where $\theta = (\mu, \lambda)$, $X$ = observed data, $W$ = complete set, $I_X(\theta)$ = observed information, $I_W(\theta)$ = complete information, and $I_{W|X}(\theta)$ = the information missing. For $\alpha \neq 1$, the likelihood function $Lc$ of the APED for the complete data is

$$Lc = \prod_{i=1}^{n} f(x_i) = \lambda^n e^{-\lambda \sum_{i=1}^{n}(x_i - \mu)} \alpha^{\sum_{i=1}^{n}(1 - e^{-\lambda(x_i - \mu)})}. \qquad (28)$$

The log-likelihood function $\ln Lc$ of the APED for the complete data is

$$\ln Lc = n \ln(\lambda) - \lambda \sum_{i=1}^{n}(x_i - \mu) + \ln(\alpha)\sum_{i=1}^{n}(1 - e^{-\lambda(x_i - \mu)}). \qquad (29)$$

The second partial derivatives of $\ln Lc$ are

$$\frac{\partial^2 \ln Lc}{\partial \mu^2} = -\lambda^2 \ln(\alpha)\sum_{i=1}^{n} e^{-\lambda(x_i - \mu)}$$

$$\frac{\partial^2 \ln Lc}{\partial \mu \partial \lambda} = n - \ln(\alpha)\sum_{i=1}^{n} e^{-\lambda(x_i - \mu)} + \lambda \ln(\alpha)\sum_{i=1}^{n}(x_i - \mu)e^{-\lambda(x_i - \mu)}$$

$$\frac{\partial^2 \ln Lc}{\partial \lambda^2} = -\frac{n}{\lambda^2} - \ln(\alpha)\sum_{i=1}^{n}(x_i - \mu)^2 e^{-\lambda(x_i - \mu)}$$

The expected values of the second derivatives are

$$\begin{cases} E\left[\dfrac{\partial^2 \ln L_c}{\partial \mu^2}\right] = -\lambda^2 \ln(\alpha)\sum_{i=1}^{n} E\left[e^{-\lambda(x_i - \mu)}\right], \\[2mm] E\left[\dfrac{\partial^2 \ln L_c}{\partial \mu \partial \lambda}\right] = n - \ln(\alpha)\sum_{i=1}^{n} E\left[e^{-\lambda(x_i - \mu)}\right] + \lambda \ln(\alpha)\sum_{i=1}^{n} E\left[(x_i - \mu)e^{-\lambda(x_i - \mu)}\right], \\[2mm] E\left[\dfrac{\partial^2 \ln L_c}{\partial \lambda^2}\right] = -\dfrac{n}{\lambda^2} - \ln(\alpha)\sum_{i=1}^{n} E\left[(x_i - \mu)^2 e^{-\lambda(x_i - \mu)}\right], \end{cases}$$

Then, the complete information becomes

$$I_W(\theta) = -E\left[\frac{\partial^2 \ln L_c}{\partial \theta^2}\right] = (-1)\begin{bmatrix} E\left[\dfrac{\partial^2 \ln L_c}{\partial \mu^2}\right] & E\left[\dfrac{\partial^2 \ln L_c}{\partial \mu \partial \lambda}\right] \\[2mm] E\left[\dfrac{\partial^2 \ln L_c}{\partial \mu \partial \lambda}\right] & E\left[\dfrac{\partial^2 \ln L_c}{\partial \lambda^2}\right] \end{bmatrix}, \tag{30}$$

where

$$\begin{aligned} E[e^{-\lambda(x_i-\mu)}] &= \frac{\lambda \ln \alpha}{\alpha - 1}\int_0^\infty e^{-2\lambda(x_i-\mu)}\alpha^{(1-e^{-\lambda(x-\mu)})}dx \\[2mm] &= \frac{\ln \alpha}{\alpha - 1}\int_0^{e^{\lambda\mu}} u \times \alpha^{(1-u)}du = \frac{\alpha^{1-e^{\lambda\mu}}[\alpha^{e^{\lambda\mu}}-1-e^{\lambda\mu}\ln\alpha]}{(\alpha - 1)(\ln\alpha)}, \end{aligned} \tag{31}$$

$$\begin{aligned} E[(x_i-\mu)e^{-\lambda(x_i-\mu)}] &= \frac{\lambda \ln \alpha}{\alpha - 1}\int_0^\infty (x_i-\mu)e^{-2\lambda(x_i-\mu)}\alpha^{(1-e^{-\lambda(x-\mu)})}dx \\[2mm] &= \frac{-\ln\alpha}{\lambda(\alpha-1)}\int_0^{e^{\lambda\mu}} u \ln(u)\alpha^{(1-u)}du, \end{aligned} \tag{32}$$

and

$$\begin{aligned} E[(x_i-\mu)^2 e^{-\lambda(x_i-\mu)}] &= \frac{\lambda \ln \alpha}{\alpha - 1}\int_0^\infty (x_i-\mu)^2 e^{-2\lambda(x_i-\mu)}\alpha^{(1-e^{-\lambda(x-\mu)})}dx \\[2mm] &= \frac{\ln\alpha}{\lambda^2(\alpha-1)}\int_0^{e^{\lambda\mu}} u \ln^2(u)\alpha^{(1-u)}du. \end{aligned} \tag{33}$$

The FIM of the censored data can be given as

$$I_{W|X}(\theta) = (n-k)E\left[-\frac{\partial^2 \ln f_{Z|X}(Z|X,\theta)}{\partial \theta^2}\right] = (n-k)\begin{bmatrix} b_{11} & b_{12} \\ b_{21} & b_{22} \end{bmatrix}, \tag{34}$$

where

$$\begin{cases} b_{11} = \lambda^2 \ln(\alpha)E[e^{-\lambda(z-\mu)}] + \dfrac{\lambda^2 \ln(\alpha)e^{-\lambda(c-\mu)}}{(-1+\alpha^{e^{-\lambda(c-\mu)}})} - \dfrac{\lambda^2 \ln^2(\alpha)e^{-2\lambda(c-\mu)}\alpha^{e^{-\lambda(c-\mu)}}}{(-1+\alpha^{e^{-\lambda(c-\mu)}})^2}, \\[3mm] b_{12} = b_{21} = \dfrac{\lambda(c-\mu)e^{-2\lambda(c-\mu)}\alpha^{e^{-\lambda(c-\mu)}}\ln^2(\alpha)}{(-1+\alpha^{e^{-\lambda(c-\mu)}})^2} + \dfrac{(1-\lambda c+\lambda\mu)e^{-\lambda(c-\mu)}\ln(\alpha)}{(-1+\alpha^{e^{-\lambda(c-\mu)}})} \\[3mm] \quad -1 + \ln(\alpha)\{(1+\lambda\mu)E[e^{-\lambda(z-\mu)}] - \lambda E[ze^{-\lambda(z-\mu)}]\}, \\[3mm] b_{22} = \dfrac{1}{\lambda^2} - \dfrac{(c-\mu)^2 e^{-2\lambda(c-\mu)}\alpha^{e^{-\lambda(c-\mu)}}\ln^2(\alpha)}{(-1+\alpha^{e^{-\lambda(c-\mu)}})^2} + \ln(\alpha)E\left[(z-\mu)^2 e^{-\lambda(z-\mu)}\right] \\[3mm] \quad -\dfrac{(c-\mu)^2 e^{-\lambda(c-\mu)}\ln(\alpha)}{(-1+\alpha^{e^{-\lambda(c-\mu)}})}, \end{cases}$$

$$E[e^{-\lambda(z-\mu)}] = \frac{\ln(\alpha)\displaystyle\int_0^{e^{-\lambda(c-\mu)}} u\alpha^{1-u}du}{(\alpha-1)(1-F(c,\mu,\lambda))} = \frac{\alpha^{1-e^{-\lambda(c-\mu)}}\{\alpha^{e^{-\lambda(c-\mu)}}-1-\ln(\alpha)e^{-\lambda(c-\mu)}\}}{(1-F(c,\mu,\lambda))(\alpha-1)\ln(\alpha)}$$

$$E[ze^{-\lambda(z-\mu)}] = \frac{\ln(\alpha)}{(\alpha-1)(1-F(c,\mu,\lambda))}\int_0^{e^{-\lambda(c-\mu)}}\{\mu-\frac{1}{\lambda}\ln(u)\}u\alpha^{1-u}du$$

$$= \frac{\mu\alpha^{1-e^{-\lambda(c-\mu)}}\{\alpha^{e^{-\lambda(c-\mu)}}-1-\ln(\alpha)e^{-\lambda(c-\mu)}\}}{(\alpha-1)\ln(\alpha)(1-F(c,\mu,\lambda))} - \frac{\ln(\alpha)\displaystyle\int_0^{e^{-\lambda(c-\mu)}} u\ln(u)\alpha^{1-u}du}{\lambda(\alpha-1)(1-F(c,\mu,\lambda))}$$

$$E\big[(z-\mu)^2e^{-\lambda(z-\mu)}\big] = \frac{\ln(\alpha)}{\lambda^2(\alpha-1)(1-F(c,\mu,\lambda))}\int_0^{e^{-\lambda(c-\mu)}} u\ln^2(u)\alpha^{1-u}du$$

To obtain the variance–covariance matrix of $\hat{\mu}$ and $\hat{\lambda}$, one can invert the observed information matrix as

$$\begin{bmatrix} Var(\hat{\mu}_{EMA}) & Cov(\hat{\mu}_{EMA},\hat{\lambda}_{EMA}) \\ Cov(\hat{\mu}_{EMA},\hat{\lambda}_{EMA}) & Var(\hat{\lambda}_{EMA}) \end{bmatrix} = I_X^{-1}(\theta) = [I_W(\theta) - I_{W|X}(\theta)]^{-1}. \tag{35}$$

The approximate $100(1-\gamma)$% confidence intervals for $\hat{\mu}$ and $\hat{\lambda}$ are

$$\hat{\mu}_{EMA} \pm Z_{\frac{\gamma}{2}}\sqrt{\widehat{Var}(\hat{\mu}_{EMA})} \text{ and } \hat{\lambda}_{EMA} \pm Z_{\frac{\gamma}{2}}\sqrt{\widehat{Var}(\hat{\lambda}_{EMA})}, \tag{36}$$

where $Z_{\frac{\gamma}{2}}$ is a standard normal variate. In addition, the $100(1-\gamma)$% ACIs of $R(t)$ and $H(t)$ are estimated using the delta method as

$$\hat{R}_{EMA}(t) \pm Z_{\frac{\gamma}{2}}\sqrt{\widehat{Var}(\hat{R}_{EMA})} \text{ and } \hat{H}_{EMA}(t) \pm Z_{\frac{\gamma}{2}}\sqrt{\widehat{Var}(\hat{H}_{EMA})}, \tag{37}$$

where

$$\widehat{Var}(\hat{R}_{EMA}) = [G_1^T I_X^{-1}(\theta)G_1]_{(\mu,\lambda)=(\hat{\mu}_{EMA},\hat{\lambda}_{EMA})}, \widehat{Var}(\hat{H}_{EMA}) = [G_2^T I_X^{-1}(\theta)G_2]_{(\mu,\lambda)=(\hat{\mu}_{EMA},\hat{\lambda}_{EMA})}, \tag{38}$$

where $G_1^T$ and $G_1^T$ are given by Eq (24).

## 4 Simulation study

This section presents Monte Carlo simulation study to estimate the performance of the MLEs of $\mu$, $\lambda$, $R(t)$, and $H(t)$ achieved by applying the NRM and EMA. The parameter values of $\mu$, $\lambda$, and $\alpha$ and sample size $n$ are required for this simulation. In this study, we used parameters values $\alpha = 10$, $\mu = 2$, and $\lambda = 1$, the sample size $n$ was set to 20, 30, and 40, and $k$ was chosen such that the observed data were 70% and 90% censored. A mission time of $t = 3.0$ was taken for the survival and failure rate functions. Hence, $R(t) = 0.6348$ and $H(t) = 1.1048$. For the point estimation methods, we compared the expected values (EVs) and mean squared errors (MSEs) of the estimators for $\mu$, $\lambda$, and the reliability and hazard functions, see Zeg et al. [26, 27]. For the interval estimation methods, the 95% confidence intervals were compared according to the average length (AL) and coverage probability (CP). For the selected options of $(n, k, T)$, the

 

**Table 1. Expected Value (EV), Mean Squared Error (MSE), Average Length (AL), and Coverage Probability (CP) of $\mu$ and $\lambda$ when $\mu = 2$, $\lambda = 1$, $\alpha = 10$ and $t = 3$ for varying $(n, k, T)$.**

| $n$ | $k$ | Parameter | | NRM | | | | EMA | | | |
|---|---|---|---|---|---|---|---|---|---|---|---|
| | | | T | EV | MSE | AL | CP | EV | MSE | AL | CP |
| 20 | 14 | $\mu$ | 1.2 | 2.1740 | 0.0522 | 1.0691 | 0.916 | 2.0542 | 0.0375 | 0.8977 | 0.942 |
| | | | 1.6 | 2.1602 | 0.0433 | 1.0568 | 0.938 | 2.0450 | 0.0326 | 0.8882 | 0.954 |
| | | | 2.0 | 2.1508 | 0.0405 | 1.0350 | 0.950 | 2.0213 | 0.0323 | 0.8875 | 0.966 |
| | | $\lambda$ | 1.2 | 1.1737 | 0.1021 | 1.0456 | 0.974 | 1.0290 | 0.0605 | 0.7487 | 0.916 |
| | | | 1.6 | 1.1554 | 0.0889 | 1.0305 | 0.970 | 1.0153 | 0.0523 | 0.7387 | 0.898 |
| | | | 2.0 | 1.1532 | 0.0859 | 1.0278 | 0.976 | 1.0123 | 0.0514 | 0.7365 | 0.914 |
| | 18 | $\mu$ | 1.2 | 2.1516 | 0.0391 | 1.1436 | 0.968 | 2.1450 | 0.0369 | 0.8011 | 0.926 |
| | | | 1.6 | 2.1462 | 0.0370 | 1.1542 | 0.962 | 2.1389 | 0.0350 | 0.8043 | 0.934 |
| | | | 2.0 | 2.1467 | 0.0366 | 1.1465 | 0.966 | 2.1388 | 0.0303 | 0.7968 | 0.932 |
| | | $\lambda$ | 1.2 | 1.1298 | 0.0707 | 0.9842 | 0.962 | 1.1131 | 0.0650 | 0.7533 | 0.942 |
| | | | 1.6 | 1.1238 | 0.0666 | 0.9791 | 0.941 | 1.1066 | 0.0609 | 0.7490 | 0.938 |
| | | | 2.0 | 1.1211 | 0.0596 | 0.9633 | 0.968 | 1.1016 | 0.0504 | 0.7423 | 0.954 |
| 30 | 21 | $\mu$ | 1.2 | 2.1245 | 0.0277 | 0.9685 | 0.972 | 2.1090 | 0.0237 | 0.6879 | 0.940 |
| | | | 1.6 | 2.1036 | 0.0212 | 0.9573 | 0.958 | 2.0869 | 0.0191 | 0.6868 | 0.942 |
| | | | 2.0 | 2.1010 | 0.0206 | 0.9512 | 0.962 | 2.0885 | 0.0174 | 0.6785 | 0.954 |
| | | $\lambda$ | 1.2 | 1.0843 | 0.0395 | 0.7921 | 0.943 | 1.0453 | 0.0322 | 0.5942 | 0.934 |
| | | | 1.6 | 1.0835 | 0.0354 | 0.7901 | 0.932 | 1.0442 | 0.0286 | 0.5936 | 0.952 |
| | | | 2.0 | 1.0786 | 0.0316 | 0.7808 | 0.944 | 1.0393 | 0.0251 | 0.5908 | 0.956 |
| | 27 | $\mu$ | 1.2 | 2.1076 | 0.0214 | 0.9625 | 0.964 | 2.1057 | 0.0213 | 0.6746 | 0.956 |
| | | | 1.6 | 2.1031 | 0.0195 | 0.9564 | 0.959 | 2.1006 | 0.0190 | 0.6719 | 0.960 |
| | | | 2.0 | 2.0631 | 0.0164 | 0.9532 | 0.956 | 2.0983 | 0.0149 | 0.6648 | 0.953 |
| | | $\lambda$ | 1.2 | 1.0753 | 0.0347 | 0.7808 | 0.978 | 1.0323 | 0.0308 | 0.5869 | 0.936 |
| | | | 1.6 | 1.0712 | 0.0316 | 0.7802 | 0.965 | 1.0316 | 0.0267 | 0.5821 | 0.948 |
| | | | 2.0 | 1.0712 | 0.0316 | 0.7802 | 0.952 | 1.0243 | 0.0246 | 0.5769 | 0.951 |
| 40 | 28 | $\mu$ | 1.2 | 2.0958 | 0.0158 | 0.8067 | 0.942 | 2.0195 | 0.0128 | 0.6358 | 0.947 |
| | | | 1.6 | 2.0885 | 0.0141 | 0.8049 | 0.968 | 2.0124 | 0.0126 | 0.6305 | 0.953 |
| | | | 2.0 | 2.0854 | 0.0135 | 0.8078 | 0.964 | 1.9995 | 0.0121 | 0.6267 | 0.955 |
| | | $\lambda$ | 1.2 | 1.0868 | 0.0333 | 0.7010 | 0.962 | 0.9769 | 0.0232 | 0.5021 | 0.942 |
| | | | 1.6 | 1.0950 | 0.0329 | 0.7091 | 0.958 | 0.9844 | 0.0218 | 0.5016 | 0.949 |
| | | | 2.0 | 1.0737 | 0.0313 | 0.6874 | 0.966 | 0.9592 | 0.0228 | 0.4930 | 0.961 |
| | 36 | $\mu$ | 1.2 | 2.0866 | 0.0132 | 0.8974 | 0.944 | 2.0855 | 0.0127 | 0.5894 | 0.954 |
| | | | 1.6 | 2.0830 | 0.0124 | 0.8847 | 0.958 | 2.0813 | 0.0120 | 0.5892 | 0.960 |
| | | | 2.0 | 2.0812 | 0.0121 | 0.8806 | 0.942 | 2.0805 | 0.0119 | 0.5860 | 0.954 |
| | | $\lambda$ | 1.2 | 1.0592 | 0.0244 | 0.6747 | 0.962 | 1.0468 | 0.0225 | 0.5008 | 0.934 |
| | | | 1.6 | 1.0584 | 0.0220 | 0.6699 | 0.966 | 1.0458 | 0.0203 | 0.5003 | 0.949 |
| | | | 2.0 | 1.0640 | 0.0201 | 0.6659 | 0.960 | 1.0516 | 0.0200 | 0.4931 | 0.948 |

MLEs of $\mu$, $\lambda$, $R(t)$ and $H(t)$ were obtained using the NRM and EMA in 1000 replications. The results are reported in Tables 1 and 2.

From these results the following conclusions can be drawn.

(i). When the number of failures $k$ is fixed and sample size $n$ increases, the MSEs and width of the 95% confidence intervals of the MLEs computed using both the EMA and NRM decrease. Therefore, the MLE process performs well in terms of estimating the parameters of APED. Moreover, the expected values are close to the true values.

**Table 2. Expected Value (EV), Mean Squared Error (MSE), Average Length (AL), and Coverage Probability (CP) of $R(t)$ and $H(t)$ when $\mu = 2$, $\lambda = 1$, $\alpha = 10$ and $t = 3$ for varying $(n, k, T)$.**

| $n$ | $k$ | Parameter | | NRM | | | | EMA | | | |
|---|---|---|---|---|---|---|---|---|---|---|---|
| | | | T | EV | MSE | AL | CP | EV | MSE | AL | CP |
| 20 | 15 | $R(t)$ | 1.2 | 0.6529 | 0.0096 | 0.4261 | 0.962 | 0.6549 | 0.0079 | 0.3864 | 0.964 |
| | | | 1.6 | 0.6521 | 0.0084 | 0.4258 | 0.954 | 0.6558 | 0.0071 | 0.3858 | 0.947 |
| | | | 2.0 | 0.6461 | 0.0082 | 0.4251 | 0.962 | 0.6476 | 0.0066 | 0.3816 | 0.954 |
| | | $H(t)$ | 1.2 | 1.4068 | 0.2773 | 2.1713 | 0.968 | 1.2318 | 0.1404 | 1.4284 | 0.968 |
| | | | 1.6 | 1.3673 | 0.1985 | 2.0733 | 0.964 | 1.2153 | 0.1286 | 1.3969 | 0.950 |
| | | | 2.0 | 1.3452 | 0.1697 | 1.9745 | 0.934 | 1.1766 | 0.0899 | 1.3217 | 0.967 |
| | 18 | $R(t)$ | 1.2 | 0.6556 | 0.0080 | 0.4059 | 0.974 | 0.6583 | 0.0079 | 0.3757 | 0.950 |
| | | | 1.6 | 0.6558 | 0.0071 | 0.4031 | 0.970 | 0.6584 | 0.0070 | 0.3723 | 0.964 |
| | | | 2.0 | 0.6526 | 0.0069 | 0.4017 | 0.971 | 0.6550 | 0.0067 | 0.3707 | 0.962 |
| | | $H(t)$ | 1.2 | 1.3503 | 0.1656 | 2.1350 | 0.968 | 1.3406 | 0.1554 | 1.5394 | 0.957 |
| | | | 1.6 | 1.3426 | 0.1608 | 2.1110 | 0.998 | 1.3323 | 0.1511 | 1.5200 | 0.996 |
| | | | 2.0 | 1.3416 | 0.1530 | 2.1065 | 0.991 | 1.3305 | 0.1502 | 1.5144 | 0.983 |
| 30 | 21 | $R(t)$ | 1.2 | 0.6579 | 0.0055 | 0.3392 | 0.974 | 0.6652 | 0.0054 | 0.3095 | 0.968 |
| | | | 1.6 | 0.6485 | 0.0049 | 0.3383 | 0.982 | 0.6559 | 0.0049 | 0.3013 | 0.97 |
| | | | 2.0 | 0.6503 | 0.0047 | 0.3376 | 0.978 | 0.6576 | 0.0047 | 0.2991 | 0.967 |
| | | $H(t)$ | 1.2 | 1.3014 | 0.1049 | 1.6675 | 0.969 | 1.2815 | 0.0865 | 1.1873 | 0.936 |
| | | | 1.6 | 1.2593 | 0.0747 | 1.5705 | 0.949 | 1.2423 | 0.0660 | 1.1335 | 0.958 |
| | | | 2.0 | 1.2608 | 0.0708 | 1.5617 | 0.960 | 1.2415 | 0.0564 | 1.1329 | 0.973 |
| | 27 | $R(t)$ | 1.2 | 0.6534 | 0.0046 | 0.3343 | 0.974 | 0.6574 | 0.0045 | 0.3086 | 0.966 |
| | | | 1.6 | 0.6501 | 0.0042 | 0.3317 | 0.978 | 0.6538 | 0.0042 | 0.3005 | 0.968 |
| | | | 2.0 | 0.6543 | 0.0040 | 0.3312 | 0.960 | 0.6545 | 0.0038 | 0.2985 | 0.956 |
| | | $H(t)$ | 1.2 | 1.2714 | 0.0682 | 1.6398 | 0.961 | 1.2697 | 0.0672 | 1.1545 | 0.967 |
| | | | 1.6 | 1.2564 | 0.0629 | 1.5496 | 0.965 | 1.2555 | 0.0621 | 1.1287 | 0.952 |
| | | | 2.0 | 1.2593 | 0.0585 | 1.5321 | 0.970 | 1.2531 | 0.0537 | 1.1216 | 0.972 |
| | | $R(t)$ | 1.2 | 0.6437 | 0.0037 | 0.3019 | 0.958 | 0.6552 | 0.0035 | 0.2744 | 0.959 |
| | | | 1.6 | 0.6372 | 0.0036 | 0.3025 | 0.966 | 0.6494 | 0.0033 | 0.2754 | 0.960 |
| | | | 2.0 | 0.6456 | 0.0030 | 0.3018 | 0.959 | 0.6569 | 0.0029 | 0.2744 | 0.956 |
| | | $H(t)$ | 1.2 | 1.2360 | 0.0412 | 1.2837 | 0.971 | 1.1527 | 0.0262 | 0.9970 | 0.950 |
| | | | 1.6 | 1.2204 | 0.0368 | 1.2614 | 0.958 | 1.1403 | 0.0241 | 0.9801 | 0.954 |
| | | | 2.0 | 1.2214 | 0.0365 | 1.2568 | 0.969 | 1.128 | 0.0226 | 0.9767 | 0.961 |
| | | | 2.0 | | | | | | | | |
| | | $R(t)$ | 1.2 | 0.6495 | 0.0033 | 0.2914 | 0.966 | 0.6535 | 0.0033 | 0.2689 | 0.953 |
| | | | 1.6 | 0.6480 | 0.0032 | 0.2900 | 0.955 | 0.6519 | 0.0032 | 0.2686 | 0.946 |
| | | | 2.0 | 0.6451 | 0.0031 | 0.2898 | 0.966 | 0.6494 | 0.0032 | 0.2689 | 0.950 |
| | | $H(t)$ | 1.2 | 1.2264 | 0.0341 | 1.3803 | 0.957 | 1.2274 | 0.0336 | 0.9513 | 0.956 |
| | | | 1.6 | 1.2205 | 0.0327 | 1.3529 | 0.961 | 1.2208 | 0.0323 | 0.9437 | 0.965 |
| | | | 2.0 | 1.2160 | 0.0317 | 1.3481 | 0.957 | 1.2180 | 0.0316 | 0.9389 | 0.958 |

(ii). The parameters estimation values under the EMA algorithm and their respective MSEs are smaller than those computed via the NRM.

(iii). As the sample size $n$ increases, the average length of all intervals decreases. On average, the ACIs obtained via the EMA have a shorter length, and the coverages of the confidence intervals in all cases are close to 95%.

(iv). The MSEs and the widths of the confidence intervals of the MLEs estimated by the EMA and NRM decrease as the number of failures ($k$) increases for a fixed $n$ sample size.

(v). The MSEs and the length of the ACIs for the parameters, as well as for the reliability and hazard functions, decrease for fixed $n$ and $k$ as $T$ increases.

## 5 Numerical examples

### 5.1 Simulated data analysis

Here, T-IIHC data with $n = 20$, $k = 15$, $T = 1.5$ were generated from APED with $\mu = 1.5$, $\lambda = 1$, and $\alpha = 3$. The generated data were

$$1.6308, \quad 0.65771, \quad 1.7338, \quad 1.7740, \quad 1.8706,$$
$$1.9477, \quad 2.0421, \quad 2.2302, \quad 2.3375, \quad 2.4247,$$
$$2.4512, \quad 2.5617, \quad 2.5857, \quad 2.7644, \quad 2.8357.$$

For reliability characteristics, we used mission time $t = 3$. Based on the T-IIHC sample, the MLEs using the NRM for $\mu$, $\lambda$, $R(t = 3)$ and $H(t = 3)$ were computed as follows

$$\hat{\mu}_{ML} = 1.6308, \hat{\lambda}_{ML} = 1.4015, \hat{R}_{ML}(t) = 0.2234, \hat{H}_{ML}(t) = 0.3715,$$

and the variance–covariance matrix is given by

$$\begin{bmatrix} 0.242874 & 0.279324 \\ 0.279324 & 0.426236 \end{bmatrix}.$$

Then, the 95% confidence intervals for $\mu$, $\lambda$, $R(t)$ and $H(t)$ when the NRM was used are (0.6649, 2.5967), (0.1218, 2.6810), (0.0411, 0.4057) and (0.0774, 0.6654), respectively.

Conversely, we used the EMA, as described in Sections 2 and 3, and stopped the iterative process when the difference between two consecutive iterations was less than $10^{-6}$. The MLEs for $\mu$ and $\lambda$ obtained via the EMA require 0.06 s and seven iterations to converge to $\hat{\mu}_{EMA} = 1.6308$ and $\hat{\lambda}_{EMA} = 1.3238$, and the MLEs $R(t = 3)$ and $H(t = 3)$ are $\hat{R}_{EMA}(t) = 0.3949$ and $\hat{H}_{EMA}(t) = 0.3949$. Further, the variance–covariance matrix is given by

$$\begin{bmatrix} 0.0303485 & -0.0295087 \\ -0.0295087 & 0.0770642 \end{bmatrix}.$$

Moreover, the 95% confidence intervals for $\mu$, $\lambda$, $R(t = 3)$ and $H(t = 3)$ are (1.2894, 1.9723), (0.7797, 1.8678), (0.0031, 0.4895) and (0.0656, 0.7241), respectively.

This example shows that the MLEs obtained by EMA converge to the true values of the unknown parameters $\mu$ and $\lambda$ better than those obtained by the NRM.

### 5.2 Real data example

In the experiment described in this subsection, one set of real data was used to demonstrate the applicability of the suggested method to real-life applications. The data, which represent the strength of a single carbon fiber and impregnated 1000-carbon fiber tows, measured in GPa, were taken from Bader and Priest [28]. Mahdavi and Kundu [18] reported the data of single carbon fibers tested at a gauge length of 1 mm as

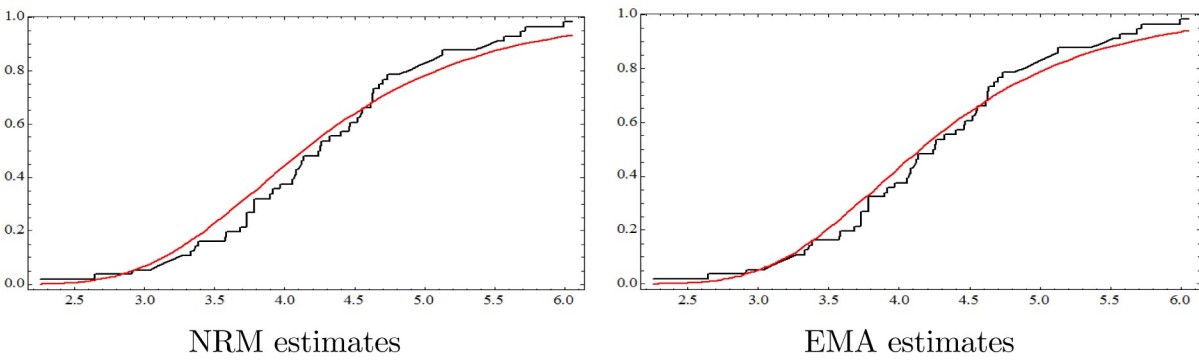

**Fig 1. Empirical and ftted survival functions of NRM and EMA estimates for the real data.**

2.247, 2.64, 2.908, 3.099, 3.126, 3.245, 3.328, 3.355, 3.383, 3.572, 3.581, 3.681, 3.726 3.727, 3.728, 3.783, 3.785, 3.786, 3.896, 3.912, 3.964, 4.05, 4.063, 4.082, 4.111, 4.118, 4.141, 4.246, 4.251, 4.262, 4.326, 4.402, 4.457, 4.466, 4.519, 4.542, 4.555, 4.614, 4.632, 4.634, 4.636, 4.678, 4.698, 4.738, 4.832, 4.924, 5.043, 5.099, 5.134, 5.359, 5.473, 5.571, 5.684, 5.721, 5.998, 6.06.

For the previous complete data, Mahdavi and Kundu [18] obtained the MLEs of $\alpha$, $\mu$, and $\lambda$, which were found to be 673.8379, 2.247, and 1.1562, respectively. They examined the validity of the APED based on the parameters $\hat{\alpha}_{ML}$, $\hat{\mu}_{ML}$ and $\hat{\lambda}_{ML}$, using the Kolmogorov–Smirnov (KS) test. They observed that the KS test results was 0.0925, and the corresponding p-value was 0.7243. Therefore, they concluded the APE model provides a good fit for the data set presented above.

Here, we estimate the values of $\mu$ and $\lambda$ when $\alpha$ is known ($\alpha = 673.8379$). First, we computed the MLEs of the unknown parameters using the NRM: $\hat{\mu}_{ML} = 2.247$ and $\hat{\lambda}_{ML} = 1.19161$. The KS distance between the fitted and empirical CDFs was 0.1117, and the associated p-value was 0.487. Therefore, according to the result of the NRMM, we cannot reject the assumption that the source of the data set is the two-parameter APED. Furthermore, the 95% confidence intervals of $\mu$ and $\lambda$ are (1.89194, 2.60206) and (0.944682, 1.43854).

We also computed the KS distance based on the EMA, where the MLEs of $\mu$ and $\lambda$ were $\hat{\mu}_{ML} = 2.36522$ and $\hat{\lambda}_{ML} = 1.2568$. The associated 95% confidence intervals were (2.0511, 2.67933) and (1.00053, 1.513), respectively. The KS distance was 0.0927 and the associated p-value was 0.7217. Based on the p-value of the KS statistic, the MLEs obtained via EMA also provide a satisfactory estimate of the data set. The empirical survival function and the fitted survival functions are drawn in Fig 1.

From the above data set, we artificially created a hybrid censored data set with $n = 56$, $k = 50$, and $T = 4$. Based on the T-IIHC sample, the MLEs obtained via the NRM and EMA for $\mu$, $\lambda$, $R(t = 4)$, and $H(t = 4)$ were computed with the associated 95% confidence intervals; see Table 3. According to Table 3, all estimates are satisfactory for this data set.

**Table 3. MLEs and 95% CIs of $\mu$, $\lambda$, $R(t)$ and $H(t)$ with $\alpha = 673.8379$ and $T = 4$, for Bader and Priest [28] data.**

| Method | NRM | | EMA | |
|---|---|---|---|---|
| | Estimate | 95% CIs | Estimate | 95% CIs |
| $\mu$ | 2.2470 | (1.8891, 2.6050) | 2.0746 | (1.6980, 2.4511) |
| $\lambda$ | 1.1691 | (0.9210, 1.4172) | 1.0718 | (0.8437, 1.2999) |
| $R(t = 4)$ | 0.5687 | (0.4632, 0.6742) | 0.5635 | (0.4573, 0.6697) |
| $H(t = 4)$ | 0.9842 | (0.7027, 1.2657) | 0.8894 | (0.6452, 1.1336) |

## 6 Conclusion

In this article, statistical inference of T-IIHC data from an APED was described. The MLE method cannot be derived analytically; therefore, the EMA and NRM were conducted to compute the considered parameters. A simulation study was performed to assess the performance of the different schemes for the APED in estimated and real data. In the simulation study, we noted that both the EMA and the NRM produced satisfactory results, but the EMA provided better estimates. Based on the T-IIHC sample, the MLEs obtained via the NRM and EMA for $\mu$, $\lambda$, $R(t = 4)$ and $H(t = 4)$ were computed, along with the associated 95% confidence intervals, and we can state that all considered estimates are satisfactory for the real data set. Its deserve to study in future the inferences on APE parameters under a balanced two-sample type-II progressive censoring scheme.

## Author Contributions

**Conceptualization:** Essam A. Ahmed.

**Data curation:** Mukhtar M. Salah, M. El-Morshedy.

**Formal analysis:** Mukhtar M. Salah, M. El-Morshedy.

**Methodology:** Mukhtar M. Salah, Essam A. Ahmed, M. S. Eliwa.

**Project administration:** Mukhtar M. Salah.

**Software:** Essam A. Ahmed, M. S. Eliwa.

**Writing – review & editing:** Ziyad A. Alhussain, Hanan Haj Ahmed.

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
