## [Decision Letter · Decision Letter 0]

28 Sep 2020

PONE-D-20-25821

Statistical Inferences For Type-II Hybrid Censoring Data From Alpha Power Exponential Distribution

PLOS ONE

Dear Dr. A. Ahmed,

Thank you for submitting your manuscript to PLOS ONE. After careful consideration, we feel that it has merit but does not fully meet PLOS ONE’s publication criteria as it currently stands. Therefore, we invite you to submit a revised version of the manuscript that addresses the points raised during the review process.

We look forward to receiving your revised manuscript.

Kind regards,

Feng Chen

Academic Editor

PLOS ONE

Journal Requirements:

5. Please update your submission to use the PLOS LaTeX template. The template and more information on our requirements for LaTeX submissions can be found at http://journals.plos.org/plosone/s/latex.

6.  Thank you for stating the following financial disclosure:

 "The funders had no role in study design, data collection and analysis, decision to publish, or preparation of the manuscript.".

Reviewers' comments:

Reviewer's Responses to Questions

**Comments to the Author**

1. Is the manuscript technically sound, and do the data support the conclusions?

Reviewer #1: Yes

Reviewer #2: Partly

2. Has the statistical analysis been performed appropriately and rigorously? 

Reviewer #1: Yes

Reviewer #2: Yes

3. Have the authors made all data underlying the findings in their manuscript fully available?

Reviewer #1: Yes

Reviewer #2: Yes

4. Is the manuscript presented in an intelligible fashion and written in standard English?

Reviewer #1: Yes

Reviewer #2: Yes

5. Review Comments to the Author

Reviewer #1: This study proposes a method for computing estimates for the location parameter and scale parameter with fixed shape parameter of the alpha power exponential distribution (APED) under type-II hybrid censored. The performance of the proposed estimation method is demonstrated by a comparison with other alternatives using a simulation study and real-world data. The paper is generally well organized and written. A minor suggestion is that more references on the model comparison criteria (such as MSE) should be acknowledged, such as:

A multivariate random parameters Tobit model for analyzing highway crash rate by injury severity. Accident Analysis and Prevention, 2017, 99: 184-191.

Jointly modeling area-level crash rates by severity: A Bayesian multivariate random-parameters spatio-temporal Tobit regression. Transportmetrica A: Transport Science, 2019, 15(2): 1867-1884.

Spatial joint analysis for zonal daytime and nighttime crash frequencies using a Bayesian bivariate conditional autoregressive model. Journal of Transportation Safety and Security, 2020, 12(4): 566-585.

Besides, some directions for future research are suggested to draw in the Conclusion Section.

Reviewer #2: The topic of this paper is interesting. The methods sound. The results are meaningful and useful. There is one suggestion to improve this paper.

1. Some related references about maximum likelihood estimations could be added.

[1] Investigating the Differences of Single- and Multi-vehicle Accident Probability Using Mixed Logit Model, Journal of Advanced Transportation, 2018, UNSP 2702360.

[2] Analysis of hourly crash likelihood using unbalanced panel data mixed logit model and real-time driving environmental big data. 2018, JOURNAL OF SAFETY RESEARCH. 65: 153-159.

[3] Injury severities of truck drivers in single- and multi-vehicle accidents on rural highway, Accident Analysis and Prevention, 2011, 43(5), 1677-1688.

6. PLOS authors have the option to publish the peer review history of their article (what does this mean?). If published, this will include your full peer review and any attached files.

Reviewer #1: No

Reviewer #2: No

---

## [Author Response · Author response to Decision Letter 0]

21 Nov 2020

Authors response to reviewer comments on article “Statistical Inferences for Type-II Hybrid Censoring Data from the Alpha Power Exponential Distribution”. We would like to thank the reviewers for their great effort to make this work more complete. We have carefully read all of the comments and have made necessary modifications to our revised manuscript. We believe the presentation of the paper has improved, thanks to your comments and contributions. We have done all the changes in the revised draft. The reviewers’ comments and our respective responses are listed below.

No. Remarks of Reviewer#1 Response from authors Modifications Done

1 Reference “ A multivariate random parameters Tobit model for analyzing highway crash rate by injury severity. Accident Analysis and Prevention, 2017, 99: 184-191”.

 Added 

2 Reference “Spatial joint analysis for zonal daytime and nighttime crash frequencies using a Bayesian bivariate conditional autoregressive model. Journal of Transportation Safety and Security, 2020, 12(4): 566-585.” Added 

3 Some directions for future research are suggested to draw in the Conclusion Section.

 Its deserve to study in future the inferences on APE parameters under a balanced two-sample type-II progressive censoring scheme. Added to conclusion 

No. Remarks of Reviewer#2 Response from authors Modifications Done

1 Reference “Investigating the Differences of Single- and Multi-vehicle Accident Probability Using Mixed Logit Model, Journal of Advanced Transportation, 2018, UNSP 2702360.

 Added

---

## [Decision Letter · Decision Letter 1]

8 Dec 2020

Statistical Inferences For Type-II Hybrid Censoring Data From Alpha Power Exponential Distribution

PONE-D-20-25821R1

Dear Dr. A. Ahmed,

We’re pleased to inform you that your manuscript has been judged scientifically suitable for publication and will be formally accepted for publication once it meets all outstanding technical requirements.

Kind regards,

Feng Chen

Academic Editor

PLOS ONE

Additional Editor Comments (optional):

Reviewers' comments:

Reviewer's Responses to Questions

**Comments to the Author**

1. If the authors have adequately addressed your comments raised in a previous round of review and you feel that this manuscript is now acceptable for publication, you may indicate that here to bypass the “Comments to the Author” section, enter your conflict of interest statement in the “Confidential to Editor” section, and submit your "Accept" recommendation.

Reviewer #1: All comments have been addressed

Reviewer #2: All comments have been addressed

2. Is the manuscript technically sound, and do the data support the conclusions?

Reviewer #1: (No Response)

Reviewer #2: Yes

3. Has the statistical analysis been performed appropriately and rigorously? 

Reviewer #1: (No Response)

Reviewer #2: Yes

4. Have the authors made all data underlying the findings in their manuscript fully available?

Reviewer #1: (No Response)

Reviewer #2: Yes

5. Is the manuscript presented in an intelligible fashion and written in standard English?

Reviewer #1: (No Response)

Reviewer #2: Yes

6. Review Comments to the Author

Reviewer #1: (No Response)

Reviewer #2: (No Response)

7. PLOS authors have the option to publish the peer review history of their article (what does this mean?). If published, this will include your full peer review and any attached files.

Reviewer #1: No

Reviewer #2: No

---

## [Editor Report · Acceptance letter]

21 Dec 2020

PONE-D-20-25821R1 

Statistical Inferences for Type-II Hybrid Censoring Data from the Alpha Power Exponential Distribution 

Dear Dr. A. Ahmed:

I'm pleased to inform you that your manuscript has been deemed suitable for publication in PLOS ONE. Congratulations! Your manuscript is now with our production department. 

Kind regards, 

on behalf of

Dr. Feng Chen 

Academic Editor

PLOS ONE